# Validation of the German Life-Space Assessment (LSA-D): cross-sectional validation study in urban and rural community-dwelling older adults

Sandra Angelika Mümken ,[1] Paul Gellert ,[1] Malte Stollwerck,[1] Julie Lorraine O'Sullivan,[1] Joern Kiselev[2]

¹Institute of Medical Sociology and Rehabilitation Science, Charité University Medicine Berlin, Berlin, Germany
²Department of Anesthesiology and Operative Intensive Care Medicine, Charité University Medicine Berlin, Berlin, Germany

**Correspondence to**
Sandra Angelika Mümken;
sandra.muemken@charite.de

## ABSTRACT

**Objectives** To develop a German version of the original University of Alabama at Birmingham Study of Aging Life-Space Assessment (LSA-D) for measurement of community mobility in older adults within the past 4 weeks and to evaluate its construct validity for urban and rural populations of older adults.

**Design** Cross-sectional validation study.

**Setting** Two study centres in urban and rural German outpatient hospital settings.

**Participants** In total, N=83 community-dwelling older adults were recruited (n=40 from urban and n=43 from rural areas; mean age was 78.5 years (SD=5.4); 49.4% men).

**Primary and secondary outcome measures** The final version of the translated LSA-D was related to limitations in activities and instrumental activities of daily living (ADL/iADL) as primary outcome measure (primary hypothesis); and with sociodemographic factors, functional mobility, self-rated health, balance confidence and history of falls as secondary outcome measures to obtain construct validity. Further descriptive measurements of health included hand grip strength, screening of cognitive function, comorbidities and use of transportation. To assess construct validity, correlations between LSA-D and the primary and secondary outcome measures were examined for the total sample, and urban and rural subsamples using bivariate regression and multiple adjusted regression models. Descriptive analyses of LSA-D included different scoring methods for each region. All parameters were estimated using non-parametric bootstrapping procedure.

**Results** In the multiple adjusted model for the total sample, number of ADL/iADL limitations (β=−0.26; 95% CI=−0.42 to −0.08), Timed Up and Go Test (β=−0.37; 95% CI=−0.68 to −0.14), shared living arrangements (β=0.22; 95% CI=0.01 to 0.44) and history of falls in the past 6 months (β=−0.22; 95% CI=−0.41 to −0.05) showed significant associations with the LSA-D composite score, while living in urban area (β=−0.19; 95% CI=−0.42 to 0.03) and male gender (β=0.15; 95% CI=−0.04 to 0.35) were not significant.

**Conclusion** The LSA-D is a valid tool for measuring life-space mobility in German community-dwelling older adults within the past 4 weeks in ambulant urban and rural settings.

**Trial registration number** DRKS00019023.

## Strengths and limitations of this study

⇒ German validation of the original University of Alabama at Birmingham Life-Space Assessment (LSA-D) for community-dwelling older adults in urban and rural settings.

⇒ Using bootstrapped bivariate and multiple adjusted regression models to attain construct validity of the LSA-D.

⇒ Recruitment had to be stopped shortly before reaching the calculated sample size due to the decision to restrict face-to-face research to contain the outbreak of the COVID-19 pandemic in March 2020.

## INTRODUCTION

Mobility, defined as 'the ability to move oneself (either independently or by using assistive devices or transportation) within environments that expand from one's home to the neighbourhood and regions beyond',[1] encompasses general independence, opportunities for social activities and freedom to experience new sites. This broad concept of mobility goes beyond the narrow conception of mobility as performance in a single functional test without considering environmental barriers and social resources although their impact on mobility has been investigated.[2 3] Therefore, the focus on single functional mobility tests can lead to misconceptions about actual mobility performance in everyday life and health practitioners may oversee possible consequences for social participation and mental health.[4]

To overcome these shortcomings of functional mobility assessments, recent studies of mobility and ageing operationalise mobility as circled areas, so-called life-spaces, that spread from the centre of one's own house and garden to the neighbourhood, the city lived in and beyond, with each life-space offering different opportunities for social

involvement, recreational activities or access to medical care.[5 6] The application of self-reported life-spaces to determine mobility of older adults was first established by May *et al* in 1985[7] and assessment of life-space mobility with standardised questionnaires was recently recommended for geriatric research.[8] Several instruments for measuring life-space mobility in specific populations and settings exist, including assessments of life-space within one's own residence for home-bound individuals[9] or residents in nursing homes and other institutions.[10 11]

One of the most frequently applied instruments for measurement of mobility in older adults using the life-space concept is the validated Life-Space Assessment (LSA) by Baker *et al*[12] as part of the University of Alabama at Birmingham (UAB) Study of Aging. The LSA provides health professionals in geriatric settings with information on availability of environmental and social resources as an outcome of mobility assessment and gives them a more comprehensive picture of the patient's needs.

The importance of the LSA for clinical practice has been shown in various studies. Kennedy *et al*[13] for instance found that a decline in life-space mobility over 6 months is associated with greater mortality in the following 6 months. Limitations on life-space mobility are associated with long-term mortality of older men,[14] cognitive decline,[15] fall risk,[16] frailty[17] and hospital admission in older adults with heart failure.[18] Furthermore, the concept has already been established in outpatient physical therapy with various community-dwelling neurological orthopaedic and surgery patients.[19] Additionally, psychological health factors like external control beliefs[20] and personal activity goals[21] influence life-space mobility. Therefore, the LSA can also supplement evaluation concepts in psychological research and treatment of older adults. The construct validity of the LSA was commonly tested by relating the LSA not only to activities and instrumental activities of daily living (ADLs/iADLs) but also self-rated health and fears of falling.[22–24] Moreover, as pointed out by Baker *et al*,[12] there is a need to validate the LSA for urban and rural settings. Recently published studies also indicate environmental factors, such as distance to services or quality of streets and sidewalks that can differ between urban and rural settings, that might influence life-space mobility by reducing or maximising the opportunities to move independently outdoors and participate in social activities.[25]

As part of validity testing, the LSA has been translated into multiple languages such as Chinese,[26] French,[24] Spanish,[22] Swedish[27] or Danish.[28] To date, two modified German versions for assessment of life-space mobility in specific populations of older adults exist: the LSA-CI captures life-space mobility of the past week for those with mild cognitive impairment.[23] In comparison, the LSA-IS is used in institutionalised settings where life-spaces are adapted to the living environment of care facilities and life-space mobility of the previous day is captured.[11 29] However, a validated and intercultural adapted version of the original LSA that can be administered in the context of a more general geriatric setting in the overall population of community-dwelling older adults is still missing. Therefore, we conducted a validation study of a German version of the original LSA (LSA-D) (see online supplemental file 1) in urban and rural areas.

### Aims and hypotheses

Our aim was to translate, apply and validate the LSA-D, a German version of the LSA from the UAB Study of Aging, for the population of urban and rural community-dwelling older adults. In line with the original validation of the LSA, we expected a moderate association of the LSA-D composite score with limitations on ADL/iADL as primary hypothesis.[12] As secondary hypotheses, we assumed moderate associations with sociodemographic measures,[12 22] functional mobility[23 30] self-rated health,[12 22] balance confidence and history of falls.[16 31] In a further step, we investigated the independent predictive validity of the proposed factors (limitations in ADL/iADL, sociodemographic measures, functional mobility, self-rated health, balance confidence and history of falls) assuming that the primary correlation of limitations on ADL/iADL is present even after adjustment for the other constructs. Finally, we expected the newly translated LSA-D to show patterns of similar strong associations in the urban and rural subsample.

### METHODS

### Study design

A cross-sectional study design was used with two German hospital clinics as study centres. The first study centre was an ambulant geriatric rehabilitation facility of the Havelland clinics located in a small town (16 000 inhabitants) in Brandenburg, Germany. The second centre was based at the Charité – Universitätsmedizin Berlin within the Department of Anesthesiology and Operative Intensive Care. Approvement for the study was given by the local Ethics Committee of the Charité – Universitätsmedizin-Berlin and the study was prospectively registered at the German Clinical Trials Register.

Sample size calculation was based on assumptions to find a moderate to strong association of $\beta/r = -0.40$[12] between the LSA-D composite score and limitations on ADL/iADL (ie, primary hypothesis), sociodemographic measures, functional mobility, self-rated health and balance confidence in all observed populations. For testing of the primary hypothesis, 92 participants or 46 subjects per setting (ie, urban/rural) were required. This was based on the following assumptions: an effect size of Pearson's correlation coefficient or standardised $\beta$ coefficient of $r/\beta = -0.40$ (p=$-0.40$ in the population) was assumed in reference to the association between the LSA composite score (LS-C) and limitations on ADL/iADL found in the original validation study of LSA.[12] The power calculation with GPower V.3.1 for bivariate correlations (test family 'exact')[32] resulted in an estimated minimum sample size of n=46 participants per setting (urban/rural) and a critical r=$-0.29$ with a type

I error rate of α=0.025 (one-sided test; corrected for multiple testing (setting urban/rural; α=0.05/2)) and a statistical power of 1−β=0.80. Recruitment commenced in November 2019 and had to be stopped in March 2020 at a sample size of 82 due to restrictions of the then starting coronavirus pandemic. A post hoc sensitivity analysis suggests that we are still able to detect effects of r=−0.30 and larger.

## Translation process

In accordance with the 2008 guidelines of the WHO,[33] forward translation into German language was separately conducted by two researchers who formulated two German versions that were discussed and then merged into one German pre-version of the LSA-D. The pre-version was given to two native English speakers for back-translation. Again, both versions of the back-translation were discussed by the two native speakers and a concerted version of the back-translation was produced. Differences between the original LSA and the concerted back-translation were discussed and reviewed with the original author of the LSA to redefine a pre-final version of the LSA-D that was pretested for understandability using cognitive interview technique among four older adults of the Charité–Universitätsmedizin Berlin to create the final LSA-D version.[34]

## Participants and recruitment

The 83 participants were divided into two groups mainly based on the size of their place of residence and taking Chistaller's theory of 'central places' into consideration that categorises living areas based on provided services and infrastructure.[35] Participants from villages (up to 5000 inhabitants) and small towns (up to 40 000 inhabitants) were classified as living in rural areas as some towns did not provide services of upscale daily needs (eg, public swimming pools). In contrast, participants who lived in the city of Berlin (3.8 million inhabitants) with its metropolitan infrastructure and services were classified as urban population.

Inclusion criteria were defined as: age of 70 years and older; ability to read and understand the questionnaire and give written informed consent. Exclusion criteria were incidences that limited mobility within the past 4 weeks, known diagnosed severe cognitive limitations or mental conditions, need of acute care and insufficient understanding of the German language. In total, 126 persons were screened for eligibility of which 28 did not fulfil the inclusion criteria and 15 were unwilling to participate. In both study centres, participants were recruited during normal healthcare routine by trained study staff and medical professionals were consulted for any uncertainty regarding the participant's eligibility. All participants received verbal and written information on the study and were given time to consider participation before giving written consent.

## Measures

Selection of primary and secondary variables for determining construct validity was based on the original validation study of the UAB and other LSA validation studies from different countries.[12 22 23 31]

### Primary outcome measures

Life-space mobility was evaluated with the translated German version of the UAB LSA. The LSA consists of a questionnaire on five different life-spaces capturing six possible levels of life-space (0, mobility within the bedroom; 1, rooms inside the home besides the bedroom; 2, area outside the house; 3, neighbourhood; 4, town or city lived in; 5, outside of town or city lived in). For each level, participants were asked (a) if they went to this level in the past 4 weeks, (b) if so, how often, (c) if they needed assistive devices or special equipment to reach that level and (d) if they needed personal help to reach that level.[12] Different scoring methods can be used with the LSA either indicating the maximum attained life-space level (LS-M), life-space that can be reached independently without any further support (LS-I), reachable life-space with possible use of equipment but without personal help (LS-E), dichotomised life-space (LS-D) that classifies a person's mobility into the ability to travel beyond the borders of their self-perceived neighbourhood and the composite score (LS-C) that summarises the attained LS level, needed equipment or personal support and frequency of visits. The LS-C score ranges from 0 to 120 points with higher scores indicating better mobility. As the LS-C score has shown a good sensitivity regarding change over time, it is frequently applied in longitudinal studies.[36 37] In cross-sectional studies, LS-I and LS-D are additional scores for describing actual mobility and associations with other health factors.[12]

Limitations on ADL/iADL were investigated using questions from the 'Survey of Aging and Retirement in Europe'.[38] Participants were asked whether they had difficulties due to physical, emotional or cognitive problems to perform 15 activities like dressing, gardening, using a map or making a telephone call. Binary response options for each activity were yes or no. Subsequently, a sum score of limitations on ADL/iADL activities was calculated ranging from 0 to 15. Higher scores indicate more functional impairments.

### Secondary outcome measures

Sociodemographic factors (ie, age, gender, height, weight, status of shared living arrangements), use of public transportation and driving status were assessed with a standardised questionnaire.

The 'Timed Up and Go Test' (TUG) is one of the most frequently used measures of balance and functional mobility in older adults and is a recommended tool for geriatric assessment.[39] During performance of the TUG, time (in seconds) is taken for rising up from a standardised chair, walking 3 m, turning around, walking back and sitting down again at a comfortable self-selected

speed.[40] Higher TUG times are associated with impaired mobility.[41 42]

The EuroQol Visual Analogue Scale (EQ-VAS) from the EQ-5D-5L version was used to record overall self-rated health of the day on a vertical VAS ranging from 0 points for the worst imaginable health to 100 points for the best conceivable health.[43] To measure balance confidence, we used the ABC-6 Scale that was translated into German and validated by Schott.[44] Participants were accounted to have a history of falls if they had fallen at least one time in the past 6 months using the criteria of the 'Frailty and Injuries: Cooperative Studies of Intervention Techniques' to define a fall.[45]

### Further descriptive measures of health

Hand grip strength was measured as maximum of three contractions with a hydraulic handheld dynamometer (Sahean SH5001; Changwon, South Korea) in the dominant hand and standardised sitting position.[46] We administered the Charlson Comorbidity Index as a method to categorise comorbidities (0–41 points) where scores of >5 indicate a higher mortality risk.[47] Cognitive status was assessed with the Mini-Cog screening tool where a score ranging from 0 to 5 can be achieved and a score of 0–2 is seen as an indicator for further investigation of cognitive status.[48]

### Statistical analysis

Means (M) and SDs were reported descriptively for continuous demographic variables (ie, age, height, weight) and health measures (ie, limitations on ADL/iADL, time in seconds needed to complete TUG, self-rated health and balance confidence). Gender, status of shared living arrangements, use of different transportation modes and history of falls were reported for the total and each subsample as absolute frequencies and percentage of participants. Distribution of the data was skewed; therefore, we used the non-parametric, bias corrected and accelerated (BCA) bootstrap method with 10 000 resamples and fixed random seeds that resamples the collected data with replacement to derive robust results.[49] With the BCA bootstrap method, coefficients and CIs can be estimated with good statistical power even if sample sizes are small and distribution of data is unknown or not normal. For investigating differences between urban and rural participants, the Welch test was performed as it has been recommended as a standard test for small samples.[50] To determine construct validity of the LSA-D, BCA bootstrap method and standardised z-scores (ie, that can be interpreted like β coefficients) of the included binary and continuous variables (ie, age, male gender, rural or urban residence, status of shared living arrangements, sum score of limitations on ADL/iADL activities, functional mobility with TUG, self-rated health and history of falls) were used for bivariate and multivariate regression analysis. Balance confidence (scores of the ABC-6 Scale) had to be excluded from multivariate regression because they revealed a correlation of $r=-0.72$ with TUG scores. To avoid multicollinearity, it was decided to include only the TUG score due to its importance as a physical measurement of functional mobility for assessing construct validity. All analyses were run using SPSS V.25. Microsoft Excel 2016 was used to create the figure.

## RESULTS

### Sample characteristics

For the total sample (N=83), mean age was 78.5 (SD=5.4) years and about half of the sample (n=41; 49.4%) were men. Forty-seven participants (56.6%) lived together with others in a shared living arrangement. In the past 4 weeks, 39 participants (47.0%) drove a car by themselves, 18 participants (21.7%) rode a bicycle and 34 participants (41.0%) used walking aids. On average, participants had a TUG of M=13.9 (SD=9.2) s. Score of limitations on ADL/iADL was moderate with M=7.8 (SD=6.2) and mean score of self-rated health was M=64.7 (SD=21.3).

When comparing urban with rural participants, those living in urban areas had significantly more ADL/iADL limitations (t(74.51)=−2.34; p=0.022, and comorbidities, t(57.27)=−2.44; p=0.018). Rural participants were significantly older (t(81)=2.43; p=0.017), needed more time to complete the TUG (t(70.65)=3.33; p=0.001), had less balance confidence (t(80.11)=−2.84; p=0.006) and had lower self-rated health (t(81)=−2.45; p=0.016). Concerning the utilisation of means of transportation, the percentage of participants who drove a car or a bicycle for independent mobility within the last 4 weeks did not differ significantly across regions. Characteristics of participants in total and separately for each region are presented in table 1.

### Descriptive statistics of the LSA-D

Life-space level 5, as LS-M, was reached by 60.2% of the total sample. A total of 32.5% of participants had an LS-I level of 5 while the remaining needed either equipment or personal help. For the urban subsample, 40.0% of the participants reached life-space level 5 as LS-M and 27.5% of urban participants reached life-space level 5 independently without any support (LS-I). In contrast, 79.1% of rural participants achieved LS-level 5 as LS-M and 37.2% did this independently without any support (LS-I). Figure 1 illustrates the different life-space measures among the total sample and urban/rural subsample. No significant differences between urban and rural participants were observed in LS-C (t(81)=1.00; p=0.323), LS-E (t(81)=0.57; p=0.571) and LS-D (t(80.99)=−1.95; p=0.054). Rural participants had a significantly higher LS-M (t(64.60)=3.83; p<0.001) and LS-I (t(77)=−2.00; p=0.049).

### Construct validity

For the total sample, associations from the bivariate regression analyses between the LSA-D composite score, demographic variables, functional mobility and other health measures were significant for age (β=−0.24; 95%

**Table 1** Participant characteristics

| Variable | Total (N=83) | | Urban (n=40) | | Rural (n=43) | |
|---|---|---|---|---|---|---|
| | N | % | N | % | N | % |
| Gender (male) | 41 | 49.4 | 23 | 57.5 | 18 | 41.9 |
| Status of shared living arrangements | 47 | 56.6 | 19 | 47.5 | 28 | 65.1 |
| Drove a car in past 4 weeks | 39 | 47.0 | 18 | 45.0 | 21 | 48.8 |
| Rode a bicycle in past 4 weeks | 18 | 21.7 | 9 | 22.5 | 9 | 20.9 |
| Used walking aid in past 4 weeks | 34 | 41.0 | 11 | 27.5 | 23 | 53.5 |
| History of falls past 6 months (>1) | 22 | 26.5 | 9 | 22.5 | 13 | 30.2 |
| | M | SD | M | SD | M | SD |
| Age (years) | 78.5 | 5.4 | 77.1 | 5.2 | **79.8*** | 5.2 |
| Height (cm) | 168.7 (3) | 10.5 | 170.8 | 11.2 | 166.5 (3) | 9.3 |
| Weight (kg) | 79.0 (3) | 18.8 | 79.7 | 19.8 | 78.3 (3) | 17.8 |
| Body mass index | 27.6 (3) | 5.4 | 27.1 | 5.8 | 28.1 (3) | 5.1 |
| Charlson Comorbidity Index (0–41) | 3.2 | 3.6 | **4.2*** | 4.5 | 2.3 | 2.3 |
| Hand grip strength (kg) | 25.8 (2) | 11.6 | 27.7 (1) | 11.8 | 24.0 (1) | 11.4 |
| Mini-Cog score (0–5) | 3.8 | 1.5 | 3.8 | 1.6 | 3.8 | 1.4 |
| LSA-D composite score (0–120) | 60.8 | 24.3 | 58.0 | 21.7 | 63.3 | 26.5 |
| ADL/iADL (number of limitations; 0–15) | 7.8 | 6.2 | **9.4*** | 6.7 | 6.2 | 5.4 |
| TUG (s) | 13.9 (5) | 9.2 | 10.5 (3) | 6.8 | **16.9† (2)** | 10.1 |
| Self-rated health (0–100) | 64.8 | 21.3 | 70.5 | 19.3 | **59.4*** | 21.8 |
| Balance confidence (0–100) | 57.8 | 32.7 | 67.8 | 28.4 | **48.4*** | 34.0 |

Numbers in brackets report number of missing values.
*Significant difference between subsamples (p<0.05).
†Significant difference between subsamples (p<0.001).
ADL, activities of daily living; iADL, instrumental activities of daily living; LSA-D, Life-Space Assessment-Deutsch; TUG, Timed Up and Go Test.

CI=−0.44 to −0.07; p=0.016), status of shared living arrangements (β=0.22; 95% CI=0.01 to 0.43; p=0.040) ADL/iADL (β=−0.23; 95% CI=−0.43 to −0.01; p=0.034), TUG (β=−0.47; 95% CI=−0.66 to −0.34; p<0.001), self-rated health (β=0.40; 95% CI=0.19 to 0.61; p<0.001) and history of falls (β=−0.35; 95% CI=−0.54 to −0.15; p<0.001). Male gender (β=−0.09; 95% CI=−0.31 to 0.13; p=0.407) and urban residence (β=−0.11; 95% CI=−0.33 to 0.10;

p=0.314) were not significant for the total sample. In the adjusted model, age, male gender, urban residence, status of shared living arrangements, number of limitations on ADL/iADL, TUG, self-rated health and history of falls were included into the equation in one step for the total sample. The result revealed significant associations for living status in shared living arrangements (β=0.22; 95% CI=0.01 to 0.44; p=0.045), limitations on

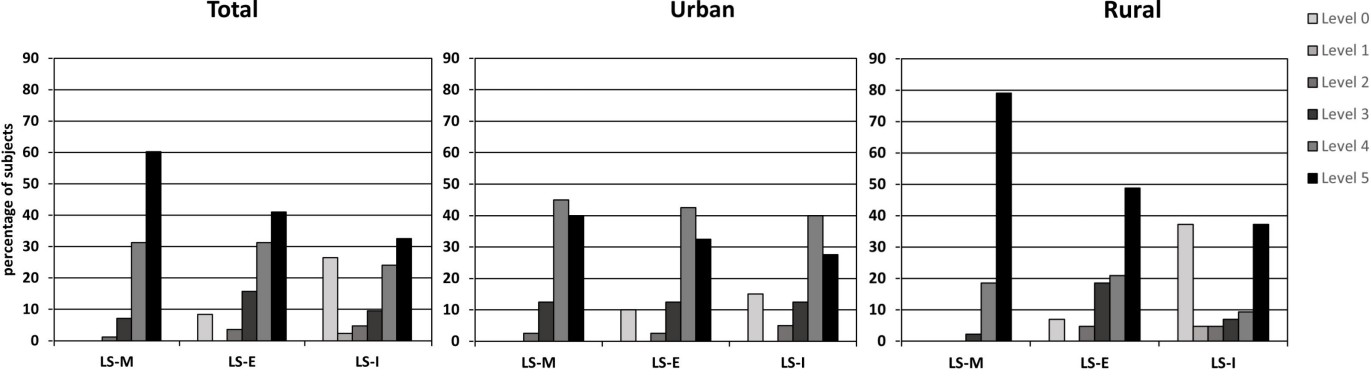

**Figure 1** Distribution of different life-space levels of the LSA-D among the total sample, urban and rural subsample. LSA-D, Life-Space Assessment-Deutsch; LS-E, life-space with equipment; LS-I, independent life-space; LS-M, maximum life-space.

**Table 2** Unadjusted and adjusted associations of sociodemographic and health factors with the LSA-D composite score (N=83)

| Variable | Bivariate unadjusted models | | | Adjusted model | | |
|---|---|---|---|---|---|---|
| | β | LL CI/UL CI | P value | β | LL CI/UL CI | P value |
| Age | −0.24 | −0.44/−0.07 | 0.016 | −0.08 | −0.32/0.12 | 0.509 |
| Gender (male) | −0.09 | −0.31/0.13 | 0.407 | 0.15 | −0.04/0.35 | 0.135 |
| Status of shared living arrangements | 0.22 | 0.01/0.43 | 0.040 | 0.22 | 0.01/0.44 | 0.045 |
| Lives in urban area | −0.11 | −0.33/0.10 | 0.314 | −0.19 | −0.42/.03 | 0.090 |
| ADL/iADL score | −0.23 | −0.43/−0.01 | 0.034 | −0.26 | −0.42/−0.08 | 0.008 |
| TUG | −0.47 | −0.66/−0.34 | <0.001 | −0.37 | −0.68/−0.14 | 0.008 |
| Self-rated health | 0.40 | 0.19/0.61 | <0.001 | 0.24 | 0.02/0.47 | 0.058 |
| Balance confidence | 0.50 | 0.33/0.67 | <0.001 | – | – | – |
| History of falls in past 6 months | −0.35 | −0.54/−0.15 | <0.001 | −0.22 | −0.41/−0.05 | 0.018 |

P<0.05 highlighted in bold.
ADL, activities of daily living; iADL, instrumental activities of daily living; LL, lower limit; LSA-D, Life-Space Assessment-Deutsch; TUG, Timed Up and Go Test; UL, upper limit.

ADL/iADL activities (β=−0.26; 95% CI=−0.42 to −0.08; p=0.008), functional mobility measured with the TUG (β=−0.37; 95% CI=−0.68 to −0.14; p=0.008) and history of falls (β=−0.22; 95% CI=−0.41 to −0.05; p=0.018). No significant associations were found for male gender (β=0.15; 95% CI=−0.04 to 0.35; p=0.135) and urban residence (β=−0.19; 95% CI=−0.42 to 0.03; p=0.090), which correspond with the bivariate model. However, in contrast to bivariate models, influence of age (β=−0.08; 95% CI=−0.32 to 0.12; p=0.509) and self-rated health (β=0.24; 95% CI=0.02 to 0.47, p=0.058) were not significant in the multivariate model. Results of bivariate and adjusted multivariate regression models are shown in table 2.

Separate bivariate regression analyses for the urban and rural region demonstrated comparable results in the urban and rural subsample for TUG urban (β=−0.48; 95% CI=−1.14 to −0.32; p=0.008) and rural (β=−0.60; 95% CI=−0.95 to −0.40; p<0.001), self-rated health urban (β=0.51; 95% CI=0.29 to 0.90; p=0.001) and rural (β=0.43; 95% CI=0.12 to 0.77; p=0.010), balance confidence urban (β=0.67; 95% CI=0.38 to 0.93; p=<0.001) and rural (β=0.54; 95% CI=0.29 to 0.81; p=0.001), and history of falls urban (β=−0.31; 95% CI=−0.59 to −0.03; p=0.030) and rural (β=−0.41; 95% CI=−0.67 to −0.11; p=0.009). Age was significant for those living in the urban region (β=−0.31; 95% CI=−0.53 to −0.09; p=0.011), but not for the rural sample (β=−0.28; 95% CI=−0.68 to 0.03; p=0.147). All other demographic variables and health measures showed no significant associations in both groups. Results are presented in table 3.

Calculations of the adjusted model for each subsample separately showed a significant regression coefficient in the urban sample for the score of ADL/iADL limitations (β=−0.23; 95% CI=−0.41 to −0.10; p=0.035) while coefficients for all other variables were not significant. For the rural population, results for status of shared living

**Table 3** Unadjusted associations of sociodemographic and health factors with LSA-D composite score for participants in urban (n=40) and rural areas (n=43)

| Variable | Urban | | | Rural | | |
|---|---|---|---|---|---|---|
| | β | LL CI/UL CI | P value | β | LL CI/UL CI | P value |
| Age | −0.31 | −0.53/−0.09 | 0.011 | −0.28 | −0.68/0.03 | 0.147 |
| Gender (male) | −0.20 | −0.43/0.06 | 0.151 | −0.04 | −0.38/0.29 | 0.826 |
| Status of shared living arrangements | 0.18 | −0.11/0.45 | 0.231 | 0.24 | −0.10/0.54 | 0.150 |
| ADL/iADL score | −0.16 | −0.39/0.11 | 0.209 | −0.28 | −0.71/0.11 | 0.208 |
| TUG | −0.48 | −1.14/−0.32 | 0.008 | −0.60 | −0.95/−0.40 | <0.001 |
| Self-rated health | 0.51 | 0.29/0.90 | 0.001 | 0.43 | 0.12/0.77 | 0.010 |
| Balance confidence | 0.67 | 0.38/0.93 | <0.001 | 0.54 | 0.29/0.81 | 0.001 |
| History of falls in past 6 months | −0.31 | −0.59/−0.03 | 0.030 | −0.41 | −0.67/−0.11 | 0.009 |

P<0.05 highlighted in bold.
ADL, activities of daily living; iADL, instrumental activities of daily living; LL, lower limit; LSA-D, Life-Space Assessment-Deutsch; TUG, Timed Up and Go Test; UL, upper limit.

arrangements (β=0.39; 95% CI=0.02 to 0.75; p=0.039) and history of falls (β=−0.42; 95% CI=−0.80 to −0.08; p=0.04) showed significance while other variables did not.

## DISCUSSION

We translated and validated the German version of the LSA in urban and rural community-dwelling older adults. In line with the original validation of the LSA,[12] moderate associations of the LSA-D composite score with limitations on ADL/iADL, living in shared living arrangements as well as with functional mobility assessed with the TUG and history of falls were found in the bivariate regression and standardised adjusted model. The standardised adjusted association of limitations on ADL/iADL with the LSA-D composite score revealed in our study is in line with findings of Baker et al[12] and Curcio et al,[22] although lower than expected. We found stronger moderate adjusted associations for functional mobility measured with the TUG. These results correspond with findings by Ullrich et al[23] who reported a moderate Pearson's correlation with the TUG when validating the modified LSA-CI to capture life-space of older adults with mild cognitive impairment during the past week.[23 30] Previous validation studies have tested their version of the LS-C against against the Short Physical Performance Battery as a physical assessment of functional mobility and found moderate to strong association.[12 27] Furthermore, our results revealed a moderate significant association for self-rated health with the LSA-D composite score in bivariate regression, which is in accordance with the original LSA validation study.[12] However, this association did not remain significant in the adjusted model. Unfortunately, balance confidence as an additional subjective health measure that showed moderate significant bivariate associations could not be included in the adjusted model due to multicollinearity. Our adjusted model confirms the importance of social resources as they can be seen in living together with others in shared living arrangements and functional mobility represented by the significant negative associations with limitations on ADL/iADL activities, time to complete the TUG and a positive history of falls.

To test our hypothesis that the LSA-D is applicable in both urban and rural living environments, we calculated separate bivariate regressions for each subsample. The associations were similarly strong for functional mobility, self-rated health, balance confidence and history of falls. Although a significant association with age was only found for the urban population and limitations on ADL/iADL were not significant in either subsample, our findings generally correspond across both subsamples. This supports our notion that the LSA-D can be used for measurement of life-space mobility during the past 4 weeks in community-dwelling older adults living in both urban and rural areas.

Contrary to our expectation, results of the adjusted model calculated separately for each subsample revealed that limitations in ADL/iADL were only significantly associated with life-space mobility in urban areas. In contrast, shared living arrangements and history of falls were the only significant adjusted factors in rural areas. One possible explanation could be that life-space mobility achieved on one's own abilities is easier to maintain in urban areas with a more pronounced infrastructure. On the contrary, a nearby social network may play a more important role for sustaining life-space mobility in rural areas where distances to services and social activities might be longer. This strengthened the importance of social resources on life-space mobility in rural areas. In this regard, Kuspinar et al[5] also found evidence for the importance of social support on life-space mobility in the Canadian Longitudinal Study of Aging. Due to the small sample size, our results should be interpreted with caution and additional studies are needed to confirm the observed differences between urban and rural community-dwelling older adults found in our study. Therefore, future studies should continue to establish a theoretical and empirical basis for urban/rural life-space mobility.

To further determine construct validity of the LSA-D, we investigated the different scoring methods in the total and both subsamples separately. The LS-M differed between urban and rural participants, with those from rural areas reporting higher LS-M than those living in urban surroundings. Older adults living in rural areas might be more dependent on leaving their village or town in order to gain access to healthcare services or to run routine errands due to the limited infrastructure often found in rural areas. Our results suggest that the LSA-D is a useful tool for capturing specific characteristics of urban and rural living environments. No group differences were found concerning the LSA-D composite score and the dichotomised LS-D scoring method. This demonstrates the ability of the LSA-D composite score and the dichotomised LS-D score to remain stable and applicable outcome measures in urban and rural living environments. Taken together, our findings demonstrate robust evidence for a good construct validity of the LSA-D.

### Strengths and limitations

A main strength of our study is that we tested construct validity of the LSA-D among urban and rural community-dwelling older adults. As mobility patterns may vary across living areas, assessments of mobility need to be valid for people living in small villages and large cities as well. Our findings revealed an urban–rural difference in LS-M and thus demonstrate that the LSA-D can detect disparities in individual mobility patterns that are related to the surrounding living area. These differences must be considered when healthcare practitioners or researchers address specific questions about independence, social support and functional mobility in different regions. Maximal life-space measured with the LS-M score is likely to vary between urban and rural areas and thus may reflect availability of different environmental resources and social support. Another strength is that we applied advanced statistical methods including non-parametric

bootstrapping procedures and multivariate regression analysis to account for confounding variables and to estimate the independent association of each of the variables with LSA-D composite score. However, there are some limitations that need to be considered. First, due to the beginning of the COVID-19 pandemic, we did not reach the planned sample size. However, post hoc sensitivity analyses revealed a sufficient statistical power. Second, although statistical power was sufficient and the focus was on community-dwelling older adults, our sample size was rather small and non-representative. Future studies should replicate our findings in a representative sample, including different subgroups of older adults and evaluate additional psychometric properties of the LSA-D such as test–retest reliability and responsiveness. Moreover, future studies should consider the overlap between life-space scoring methods of the LSA-D, social constructs and objectively derived data measured with technologies of Global Positioning System (GPS). This multimodal approach can lead to a better understanding of complex mobility patterns in older adults and associated factors.

## CONCLUSION

In conclusion, the LSA-D has shown good construct validity and can be used in the general population of community-dwelling older adults in urban and rural living environments. The use of LSA-D is recommended for geriatric healthcare practitioners of different disciplines to assess mobility in the context of social participation and health service utilisation.

**Acknowledgements** We sincerely thank Patricia Sawyer for her feedback on the pre-final, back-translated English LSA-D version. Also, we would like to thank Monica Schanzer and Rudolf Mörgeli for the back-translation of the LSA-D, all participants and all members of the study staff who are not listed as authors.

**Contributors** JK developed the first draft of the study. JK, SAM and PG planned and conducted the final study protocol. JK and SAM translated LSA into German, compiled the final version of the LSA-D, and contributed in recruiting and testing participants. SAM drafted the manuscript and conducted the major analyses. MS drafted tables and the figure. PG established the statistical analysis plan and performed the sample size and power calculation. MS assisted with data analyses and interpretation of the results. JLOS gave advice and commented on the manuscript. All authors critically reviewed the manuscript and approved the final version.

**Funding** The study was partly financed by the 'MOBILE' Project funded by the Federal Ministry of Education and Research (BMBF; ID 01GY1803) and the Institute of Medical Sociology and Rehabilitation Science, Charité–Universitätsmedizin Berlin. We acknowledge support from the German Research Foundation (DFG) and the Open Access Publication Fund of Charité–Universitätsmedizin Berlin.

**Competing interests** None declared.

**Patient consent for publication** Not required.

**Ethics approval** Ethics approval for the study was provided by the Ethics Committee of Charité–Universitätsmedizin Berlin (REC Reference EA2/124/19).

**Provenance and peer review** Not commissioned; externally peer reviewed.

**Data availability statement** Data are available upon reasonable request. Data will be made available on reasonable, methodologically sound request as soon as possible to achieve the aims of the approved proposal. Available data include individual participant data that underlie the results reported in this article after its deidentification, data dictionary and the analytical code. Data will be available at least 3 months after, and ending 5 years following article publication. Proposals should be directed at sandra.muemken@charite.de or joern.kiselev@charite.de.

**ORCID iDs**
Sandra Angelika Mümken http://orcid.org/0000-0002-0603-0855
Paul Gellert http://orcid.org/0000-0001-7492-7210

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
