## [Reviewer comments · BMJ Open]

ARTICLE DETAILS

TITLE (PROVISIONAL)	Validation of the German Life-Space Assessment LSA-D: Cross sectional validation study in urban and rural community-dwelling older adults
AUTHORS	Mümken, Sandra Angelika; Gellert, Paul; Stollwerck, Malte; O'Sullivan, Julie Lorraine; Kiselev, Joern

VERSION 1 – REVIEW

REVIEWER	Werner, C Heidelberg University, Centre for Geriatric Medicine
REVIEW RETURNED	07-Apr-2021

GENERAL COMMENTS	General comments: The manuscript presents a study on the translation and validation of a German version of the University of Alabama at Birmingham (UAB) Study of Aging Life-Space Assessment (LSA-D) in urban and rural populations of older adults. The validity of the UAB-LSA was demonstrated by analyzing bivariate and multiple associations with construct variables including (i)ADL impairments, sociodemographic variables, functional mobility, self-rated health and history of falls. Overall, the manuscript is timely, well-written, with a good introduction and study rationale, clearly presented results, and conclusions in line with the findings. I have mainly only minor comments. Abstract: - It is mentioned that “psychometric properties” are evaluated; however, only the construct validity was investigated in this study. Please be more specific here.- What was done with the secondary outcome measures? As these measures were not further addressed in the abstract, I think it is not necessary to mention them.- Post-hoc analyses were mentioned but no results were presented, thus I would suggest deleting this information in the abstract. Introduction: - “Moreover, as Baker et al.[12] pointed out, there is a need to validate the LSA in urban and rural settings.” As this is a primary aim of the manuscript, the authors should provide more information on the rationale on why there is this specific need to validate the LSA in urban and rural settings.- “...where life-spaces are adapted to the living environment of care facilities (LSA-IS) [11]” The authors could also add Hauer et al 2021 Int. J. Environ. Res. Public Health and mention that the LSA-IS captures life-space of the past day. Aims and Hypotheses:
--

	- The abbreviation UAB has already been introduced in the introduction. - For the primary hypothesis, the authors provide a reference that supports this hypothesis. However, on which references are the other hypotheses based on (moderate associations with TUG, self-rated health and history of falls; same associations in rural and urban areas)? The authors provide a lot of references in the discussion that can already be used here as a rationale for these hypotheses. - At this section, I would rather use the overall term “functional mobility” instead of using the detailed assessment test (TUG) to be consistent with the other terms “ADL limitations, self-rated health and history of falls”. At a later stage, the detailed test for assessing functional mobility (i.e. TUG) can be mentioned. Study design: - “Recruitment took place from November 2019 to February 2020 and ended in March 2020”. This sentence is a bit confusing. So the recruitment period was Nov 2019 to March 2020? Participants and recruitment: - “Participants from villages (< 5,000 inhabitants) and small towns (up to 40,000 inhabitants) were classified as living in rural areas.” Is this definition based on an established reference? - How the authors assessed the inclusion criteria for “severe cognitive limitations”? Primary outcome measures - “Life-space was evaluated with the translated German...”. Please use “life-space mobility” instead of “life-space” Secondary outcome measures - It remains a bit unclear for what these secondary outcome measures were assessed. Only for descriptive purposes of the study sample? For example, why cognitive function was not considered for construct validity testing, as it has been used in other LSA validation studies (e.g., Curcio et al., 2013; Ji et al., 2015 Arch Gerontol Geriatr, etc.). - Overall, I do not feel comfortable with the different headings “sociodemographic measures, primary outcome measures, and secondary outcome measures”. What is the difference between sociodemographic measures and secondary outcome measures? I think the primary outcome measure is the LSA and all other measures are “secondary outcomes”. From my point of view, it is rather confusing for the reader when describing primary and secondary outcomes. Statistical analysis - Why the authors decided to calculate a multivariate regression analysis? Most previous studies investigating the validity of the LSA only calculated bivariate associations. Table 1: - Please indicate the significant differences in participant characteristics between urban and rural living persons. Discussion: - “...LSA composite score against against the Short Physical Performance Battery”. Please remove one “against”. - The differences found in the adjusted models between the urban
--	--

	and rural living people should be discussed in more detail. What are possible reasons for different findings? There are no hypotheses provided for potential differences between the two living areas. Did the authors assumed differences in the results on construct validity between them? Strength and limitations: - “we tested psychometric properties.-.” Actually, only construct validity was tested, but no other psychometric property (e.g. test-retest, feasibility, responsiveness, etc.). This could also be mentioned as future research.
--	--

REVIEWER	Chilibeck, Philip University of Saskatchewan
REVIEW RETURNED	08-Apr-2021

GENERAL COMMENTS	The study involved the validation of the German version of a life space assessment tool in urban and rural community-dwelling older adults. The manuscript is relatively well-written and sample size is justified. I have a few comments: Page 5 of 28, second paragraph: “like external control believes” – should “believes” be “beliefs” here? Later same paragraph: “activities an instrumental activities of daily living (ADL/iADL)” – “an” doesn’t seem to be the correct word to use here. Please re-word. There is a question about novelty of the study – please indicate whether the LSA has been validated in other countries for urban and rural community-dwelling older adults and why a validation for the German version is justified. It is indicated participants gave consent to participate, but please also add whether the study was approved by an ethics board. Top of page 9 of 28: For the scoring “(0. Mobility within the bedroom, 1. rooms inside the home besides the bedroom, 2. area outside the house, 3. neighbourhood, 4. town or city lived in, outside of town or city lived in). Is this correct, or is there a score missing for the last category “outside of town or city lived in”? Same page, last paragraph: Please define the abbreviation “TUG” the first time you use it. Last sentence on this page “in a comfortable self-selected speed” – change “in” to “at” Page 10, first sentence: “Higher TUG times are associated with stronger mobility and ADL restrictions” – I think “stronger mobility” is incorrect here. Should this be “impaired mobility”? End of second paragraph on this page: Change “criteria’s” to “criteria” Table 1: Could the authors add a column on urban vs. rural differences for the assessed variables (i.e. p-value?). Figure: “LS-C composite life-space” is presented in the figure legend, but this does not appear in the figure.
--

VERSION 1 – AUTHOR RESPONSE

Reviewer 1

1. It is mentioned that “psychometric properties” are evaluated; however, only the construct validity was investigated in this study. Please be more specific here.

Response 1: Thanks to the reviewer for pointing this out, we agree that the term “psychometric properties” may seem misleading here. We therefore opted for use of the more specific term construct validity within the manuscript.

Action 1: We changed the term psychometric properties and used the more specific term construct validity instead in the abstract p. 2 line 14, p. 2 line 16 and throughout the entire manuscript.

Now the abstract reads as follows p. 2, line 10-20 :

Primary and secondary outcome measures

“The final version of the translated LSA-D was related with limitations in activities and instrumental activities of daily living (ADL/iADL) as primary outcome measure (primary hypothesis) and with sociodemographic factors, functional mobility, self-rated health, balance confidence and history of falls as secondary outcome measures to obtain construct validity. Further descriptive measurements of health included handgrip strength, screening of cognitive function, comorbidities and use of transportation. To assess construct validity, correlations between LSA-D and the primary and secondary outcome measures were examined for the total sample, and urban and rural subsamples using bivariate regression and multiple adjusted regression models. Descriptive analyses of LSA-D included different scoring methods for each region. All parameters were estimated using non-parametric bootstrapping procedure.”

2. What was done with the secondary outcome measures? As these measures were not further addressed in the abstract, I think it is not necessary to mention them,

#Response 2: Thank you for this comment to clarify the abstract. As it is necessary to structure the abstract by primary and secondary outcome measures to fit the guidelines of BMJ Open we decided to revise the abstract and subheadings in the article. We did this in combination with our actions to the reviewer’s comment no.13 (see comment no.13).

#Action 2: The following changes have been made in the abstract on page 2, line 10-15:

“The final version of the translated LSA-D was related with limitations in activities and instrumental activities of daily living (ADL/iADL) as primary outcome measure (primary hypothesis) and with sociodemographic factors, functional mobility, self-rated health, balance confidence and history of falls as secondary outcome measures to obtain construct validity. Further descriptive measurements of health included handgrip strength, screening of cognitive function, comorbidities and use of transportation.”

Subheadings in the measures section on p. 9-11 have been changed into:

“Primary outcome measures”; “Secondary outcome measures”; “Further descriptive measures of health”

3. Post-hoc analyses were mentioned but no results were presented, thus I would suggest deleting this information in the abstract

Response 3: Thank you for this comment. The term post-hoc analyses was wrong here. We changed the term post-hoc analyses in descriptive analyses and clarified it's use.

Action 3: We included the following changes to the abstract on page 2, line 18-20:
"Descriptive analyses of the LSA-D included different scoring methods for each region. All parameters were estimated using non-parametric bootstrapping procedure. "

4. Moreover, as Baker et al.[12] pointed out, there is a need to validate the LSA in urban and rural settings." As this is a primary aim of the manuscript, the authors should provide more information on the rationale on why there is this specific need to validate the LSA in urban and rural settings.

Response 4: Thank you for giving us the opportunity to clarify this issue. We revised and added information to the text with inclusion of one new reference (Miyashita et al. 2021). Miyashita et al. 2021 confirms the necessity to validate the LSA-D in urban and rural populations as environmental factors that may differ between urban and rural sites influence life-space mobility of community dwelling older adults

Action 4: The following changes have been made on page 5, line 14-18:
"Moreover, as pointed out by Baker et al.[12], there is a need to validate the LSA for urban and rural settings. Recently published studies also indicate environmental factors, such as distance to services or quality of streets and sidewalks, that differ between urban and rural settings might influence life-space mobility by reducing or maximising the opportunities to move independently outdoors and participate in social activities.[25] "

[12] Baker PS, Bodner EV, Allman RM. Measuring life-space mobility in community-dwelling older adults. *J Am Geriatr Soc* 2003;51:1610-4.

[25] Miyashita T, Tadaka E, Arimoto A. Cross-sectional study of individual and environmental factors associated with life-space mobility among community-dwelling independent older people. *Environmental Health and Preventive Medicine* 2021;26:9.

5. "...where life-spaces are adapted to the living environment of care facilities (LSA-IS) [11]" The authors could also add Hauer et al 2021 *Int. J. Environ. Res. Public Health* and mention that the LSA-IS captures life-space of the past day.

Response 5: Thank you for this comment. It is reasonable to describe the mentioned LSA-IS more specifically.

Action5: We revised and amended the manuscript according to your suggestion. Please find the revised sentence on page 5, line 20-25:

"To date, two modified German versions for assessment of life-space mobility in specific populations of older adults exist: the LSA-CI captures life-space mobility of the past week for those with mild cognitive impairment [23]. In comparison, the LSA-IS is used in institutionalized settings where life-spaces are adapted to the living environment of care facilities and life-space mobility of the previous day is captured.[11, 29]"

[11] Hauer K, Ullrich P, Heldmann P, et al. Validation of the interview-based life-space assessment in institutionalized settings (LSA-IS) for older persons with and without cognitive impairment. *BMC Geriatrics* 2020;20:534.

[23] Ullrich P, Werner C, Bongartz M, et al. Validation of a Modified Life-Space Assessment in Multimorbid Older Persons With Cognitive Impairment. *Gerontologist* 2019;59:e66-e75.

[29] Hauer K, Ullrich P, Heldmann P, et al. Psychometric Properties of the Proxy-Reported Life-Space Assessment in Institutionalized Settings (LSA-IS-Proxy) for Older Persons with and without Cognitive Impairment. *International journal of environmental research and public health* 2021;18:3872.

6. The abbreviation UAB has already been introduced in the introduction.

Response 6: We thank the reviewer for bringing this typo to our attention.

Action 6: We deleted the explanation of the UAB abbreviation on p.6 line 6. And now it reads as follows on p.6 line 5-6: "Our aim was to translate, apply and validate the LSA-D, a German version of the LSA from the UAB Study of Aging for the population of urban and rural community-dwelling older adults."

7. For the primary hypothesis, the authors provide a reference that supports this hypothesis. However, on which references are the other hypotheses based on (moderate associations with TUG, self-rated health and history of falls; same associations in rural and urban areas)? The authors provide a lot of references in the discussion that can already be used here as a rationale for these hypotheses.

Response 7: We thank the reviewer for this comment. Adding references to our further hypotheses provides more transparency and made our rationale clearer. We included suitable references to formulate our further hypotheses concerning the LSA-D.

Action 7: We added references from the Introduction to derive our aim and hypotheses.

The following changes have been made on p. 6, line 5-16:

"Aims and hypotheses

"Our aim was to translate, apply and validate the LSA-D, a German version of the LSA from the UAB Study of Aging for the population of urban and rural community-dwelling older adults. In line with the original validation of the LSA we expected a moderate association of the LSA-D composite score with limitations in ADL/iADL as primary hypothesis.[12] As secondary hypotheses, we assumed moderate associations with sociodemographic measures[12, 22], functional mobility[23, 30] self-rated health[12, 22], balance confidence and history of falls[16, 31]. In a further step we investigated the independent predictive validity of the proposed factors (limitations in ADL/iADL, sociodemographic measures, functional mobility, self-rated health, balance confidence and history of falls) assuming that the primary correlation of limitations in ADL/iADL is present even after adjustment for the other constructs. Finally, we expected the newly translated LSA-D to show patterns of similar strong associations in the urban and rural subsample.

[12] Baker PS, Bodner EV, Allman RM. Measuring life-space mobility in community-dwelling older adults. *J Am Geriatr Soc.* 2003 Nov;51(11):1610-4. doi: 10.1046/j.1532-5415.2003.51512.x. PMID: 14687391.

[16] Lo AX, Rundle AG, Buys D, et al. Neighborhood Disadvantage and Life-Space Mobility Are Associated with Incident Falls in Community-Dwelling Older Adults. *Journal of the American Geriatrics Society* 2016;64:2218-25.

[22] Curcio CL, Alvarado BE, Gomez F, et al. Life-Space Assessment scale to assess mobility: validation in Latin American older women and men. *Aging clinical and experimental research* 2013;25:553-60.

[23] Ullrich P, Werner C, Bongartz M, et al. Validation of a Modified Life-Space Assessment in Multimorbid Older Persons With Cognitive Impairment. *Gerontologist* 2019;59:e66-e75.

[30] Ullrich P, Eckert T, Bongartz M, et al. Life-space mobility in older persons with cognitive impairment after discharge from geriatric rehabilitation. *Archives of gerontology and geriatrics* 2019;81:192-200.

[31] Auais M, Alvarado B, Guerra R, et al. Fear of falling and its association with life-space mobility of older adults: a cross-sectional analysis using data from five international sites. *Age Ageing* 2017;46:459-65.

8. At this section, I would rather use the overall term “functional mobility” instead of using the detailed assessment test (TUG) to be consistent with the other terms “ADL limitations, self-rated health and history of falls”. At a later stage, the detailed test for assessing functional mobility (i.e. TUG) can be mentioned.

Response 8: We thank the reviewer for this helpful suggestion. Indeed, replacing the detailed description “moderate associations with Timed-Up&Go-Test (TUG)” in this section and using the term “functional mobility” instead makes the statement of hypothesizes more coherent.

Action 8: We now use the overall construct “functional mobility” instead of the measurement instrument Timed-Up&Go-Test (TUG) in the section: “Aims and Hypotheses” on p. 6 line 10 and though the entire manuscript where applicable. Please also see the abstract on comment no.7. Furthermore, as a response to the second reviewers comment we now explain the abbreviation “TUG” in the measurement section on p.10, line 7-10:

P. 6 line 8-11: “As secondary hypothesizes, we assumed moderate associations with sociodemographic measures[12, 22], functional mobility[23, 30] self-rated health[12, 22], balance confidence and history of falls[16, 31].”

P.10, line 8-10: “Secondary outcome measures

The “Timed-Up&Go-Test” TUG is one of the most frequently used measures of balance and functional mobility in older adults and is a recommended tool for geriatric assessment.[39]”

[12] Baker PS, Bodner EV, Allman RM. Measuring life-space mobility in community-dwelling older adults. *J Am Geriatr Soc.* 2003 Nov;51(11):1610-4. doi: 10.1046/j.1532-5415.2003.51512.x. PMID: 14687391.

[16] Lo AX, Rundle AG, Buys D, et al. Neighborhood Disadvantage and Life-Space Mobility Are Associated with Incident Falls in Community-Dwelling Older Adults. *Journal of the American Geriatrics Society* 2016;64:2218-25.

[22] Curcio CL, Alvarado BE, Gomez F, et al. Life-Space Assessment scale to assess mobility: validation in Latin American older women and men. *Aging clinical and experimental research* 2013;25:553-60.

[23] Ullrich P, Werner C, Bongartz M, et al. Validation of a Modified Life-Space Assessment in Multimorbid Older Persons With Cognitive Impairment. *Gerontologist* 2019;59:e66-e75.

[30] Ullrich P, Eckert T, Bongartz M, et al. Life-space mobility in older persons with cognitive impairment after discharge from geriatric rehabilitation. *Archives of gerontology and geriatrics* 2019;81:192-200.

[31] Auais M, Alvarado B, Guerra R, et al. Fear of falling and its association with life-space mobility of older adults: a cross-sectional analysis using data from five international sites. *Age Ageing* 2017;46:459-65.

[39] Turner G, Clegg A. Best practice guidelines for the management of frailty: a British Geriatrics Society, Age UK and Royal College of General Practitioners report. *Age and Ageing* 2014;43:744-7.

9. Recruitment took place from November 2019 to February 2020 and ended in March 2020“. This sentence is a bit confusing. So the recruitment period was Nov 2019 to March 2020?

Response 9: Thanks to the reviewer for noticing this inconsistency.

Action 9: We corrected the sentence in the methods section. Page 7, line 13-16:

“Recruitment commenced in November 2019 and had to be stopped in March 2020 at a sample size of 82 due to restrictions of the then starting coronavirus pandemic.”

10. “Participants from villages (< 5,000 inhabitants) and small towns (up to 40,000 inhabitants) were classified as living in rural areas.” Is this definition based on an established reference?

Response 10: We would like to thank the reviewer for pointing out that this classification needs further explanation and one reference to explain our classification transparently.

Action 10: We described the classification process in more detail and added one missing reference and the following changes have been made on p. 8 line 4-12:

“Participants and recruitment

The 83 participants were divided into two groups mainly based on the size of their place of residence and taking Chistaller’s theory of “central places” into consideration that categorizes living areas based on provided services and infrastructure [35]. Participants from villages (up to 5,000 inhabitants) and small towns (up to 40,000 inhabitants) were classified as living in rural areas as some towns missed to provide services of upscale daily needs (e.g. public swimming pools). In contrast, participants who lived in the city of Berlin (3.8 million inhabitants) with its metropolitan infrastructure and services were classified as urban population.”

[35] Einig K, Zaspel-Heisters B. Das System Zentraler Orte in Deutschland. In: Flex F, Greiving S, eds. Neuaufstellung des Zentrale-Orte Konzepts in Nordrhein-Westfalen. Hannover: Verlag des ARL 2016.

11. How the authors assessed the inclusion criteria for “severe cognitive limitations”?

Response 11: We thank the reviewer for commenting on the screening process and we would like to describe it in more detail. Patients were not included in the study if a diagnosis of dementia, or other severe cognitive or mental conditions was present. This was assessed by screening the patient’s record and if there were any uncertainties these were resolved by communicating with the responsible health professionals.

Action 11: We added further information on the screening process:

p. 8, line 14-20 now reads: “Exclusion criteria were incidences that limited mobility within the past four weeks, known diagnosed severe cognitive limitations or mental conditions, need of acute care and insufficient understanding of the German language. In total, 126 persons were screened for eligibility of which 28 did not fulfil the inclusion criteria and 15 were unwilling to participate. In both study centres, participants were recruited during normal health care routine by trained study staff and medical professionals were consulted by any uncertainty regarding the participant’s eligibility.”

12. Life-space was evaluated with the translated German...”. Please use “life-space mobility” instead of “life-space”

Response 12: We thank the reviewer for pointing out this typo. We added the word “mobility”.

Action 12: Page 9 of 28, line 6-7: Life-space mobility was evaluated with the translated German Version of the UAB Life-Space Assessment.”

13. It remains a bit unclear for what these secondary outcome measures were assessed. Only for descriptive purposes of the study sample? For example, why cognitive function was not considered for construct validity testing, as it has been used in other LSA validation studies (e.g., Curcio et al., 2013; Ji et al., 2015 Arch Gerontol Geriatr, etc.).

Response 13: Thanks to the reviewer for this question and we see the need to be more specific here. We used the Mini-Cog only for descriptive purposes.

As it was the rationale of our study to validate the LSA-D in urban and rural community-dwelling older adults in line with the original validation study of Baker et al. and we decided to focus the study design on the question whether the LSA-D is a valid tool in both settings. The Mini-Cog was only administered for descriptive purposes as it is a very brief screening tool. If we had aimed to include

cognitive status as a secondary outcome in our study, we would have opted for a more elaborate measure of cognitive function. The Leganés cognitive test (LCT) used by Cucio et al. is a screening test that covers more areas of cognitive testing (e.g. temporal orientation, spatial orientation, late memory) than the Mini-Cog, so considering cognitive function would be beyond the scope of our study. None of the participants had a recorded diagnosed dementia.

Action 13: We renamed the subheadings of the measurement section in “Primary outcome measures” (LSA-D and ADL/iADL); “Secondary outcome measures”(Sociodemographic factors, functional mobility measured with Timed-Up&Go-Test ; self-rated health from the EQ5D; Balance confidence with ABC-6 and history of falls of the past 6 month) and “Further descriptive measures of health” (handgrip strength; Charlson comorbidity index; cognitive status with the Mini-Cog). We combined this with the suggestion of the reviewers comment no. 2 on clarifying the use of secondary outcomes and comment no. 14. on the chosen subheadings. We also revised our whole manuscript and especially the discussion to make the sections clearer. We hope that we now can provide a coherent measurement description.

14. Overall, I do not feel comfortable with the different headings “sociodemographic measures, primary outcome measures, and secondary outcome measures”. What is the difference between sociodemographic measures and secondary outcome measures? I think the primary outcome measure is the LSA and all other measures are “secondary outcomes”. From my point of view, it is rather confusing for the reader when describing primary and secondary outcomes

Response 14: We would like to thank the reviewer for pointing out this important issue. The guidelines of the BMJ Open for submitting an original research article state that the primary and secondary outcomes should be addressed. However, we reorganized the structure of subheadings and arranged it in line with the journals guidelines

Action 14: Please see also #Action 2 and # Action 13. We choose to hold on to the subheadings as recommended by the guidelines of BMJ Open, but tried to clarify the differences.

15. Why the authors decided to calculate a multivariate regression analysis? Most previous studies investigating the validity of the LSA only calculated bivariate associations.

Response 15: Thanks to the reviewer for this question. in line with previous validation studies, we also calculated the bivariate regression analyses in a first step. In addition, we present findings of multivariate regression analyses to assess independent explanatory/predictive contributions of each correlated variable and to investigate possible differences between urban and rural community dwelling older adults. We hope to advance the evidence base for the association of constructs with the life-space concept and have added a sentence for this rationale in the revised manuscript.

Action 15. Aims, page 6, line 4-16:

“Aims and hypotheses

Our aim was to translate, apply and validate the LSA-D, a German version of the LSA from the UAB Study of Aging for the population of urban and rural community-dwelling older adults. In line with the original validation of the LSA we expected a moderate association of the LSA-D composite score with limitations in ADL/iADL as primary hypothesis.[12] As secondary hypotheses, we assumed moderate associations with sociodemographic measures[12, 22], functional mobility[23, 30] self-rated health[12, 22], balance confidence and history of falls[16, 31]. In a further step we investigated the independent predictive validity of the proposed factors (limitations in ADL/iADL, sociodemographic measures, functional mobility, self-rated health, balance confidence and history of falls) assuming that the primary correlation of limitations in ADL/iADL is present even after adjustment for the other constructs. Finally, we expected the newly translated LSA-D to show patterns of similar strong

associations in the urban and rural subsample.”

[12] Baker PS, Bodner EV, Allman RM. Measuring life-space mobility in community-dwelling older adults. *J Am Geriatr Soc.* 2003 Nov;51(11):1610-4. doi: 10.1046/j.1532-5415.2003.51512.x. PMID: 14687391.

[16] Lo AX, Rundle AG, Buys D, et al. Neighborhood Disadvantage and Life-Space Mobility Are Associated with Incident Falls in Community-Dwelling Older Adults. *Journal of the American Geriatrics Society* 2016;64:2218-25.

[22] Curcio CL, Alvarado BE, Gomez F, et al. Life-Space Assessment scale to assess mobility: validation in Latin American older women and men. *Aging clinical and experimental research* 2013;25:553-60.

[23] Ullrich P, Werner C, Bongartz M, et al. Validation of a Modified Life-Space Assessment in Multimorbid Older Persons With Cognitive Impairment. *Gerontologist* 2019;59:e66-e75.

[30] Ullrich P, Eckert T, Bongartz M, et al. Life-space mobility in older persons with cognitive impairment after discharge from geriatric rehabilitation. *Archives of gerontology and geriatrics* 2019;81:192-200.

[31] Auais M, Alvarado B, Guerra R, et al. Fear of falling and its association with life-space mobility of older adults: a cross-sectional analysis using data from five international sites. *Age Ageing* 2017;46:459-65.

16. : Table 1: Please indicate the significant differences in participant characteristics between urban and rural living persons

Response 16: Thank you for this comment.

Action 16: To make the differences clearer we marked significant differences in table 1, p.13 in bold letters and classified the differences in $p < .05$ and $p < .001$.

Now the notes of table 1 read as follows:

„Note: Live-Space-Assessment-Deutsch (LSA-D); Activities of Daily Living (ADL); Instrumented Activities of Daily Living (iADL); Timed-Up &Go-Test (TUG); Numbers in brackets report number of missing values;

* significant difference between subsamples ($p < .05$)

** significant difference between subsamples ($p < .001$)”

17. :”...LSA composite score against against the Short Physical Performance Battery”. Please remove one "against".

Response 17: We would like to thank the reviewer for this remark.

Action 17: We removed the surplus word “against” accordingly.

18. : The differences found in the adjusted models between the urban and rural living people should be discussed in more detail. What are possible reasons for different findings? There are no hypotheses provided for potential differences between the two living areas. Did the authors assumed differences in the results on construct validity between them?

Response 18: We thank the reviewer for shedding light on this important aspect. As a starting point, we assumed no differences in the adjusted model between urban and rural participants. Now we state this hypothesis clear in the “Aims and Hypothesis” section of our manuscript. In accordance with the reviewers comment we revised our discussion and discussed the found differences between subsamples in more detail.

Action 18:

In the revised manuscript we now state our hypothesis of no differences clearly:

On p. 6, line 15-16 it reads now: “Finally, we expected the newly translated LSA-D to show patterns of similar strong associations in the urban and rural subsample.”

Additionally, we changed the structure of our discussion and revised the abstract on found differences between urban and rural community-dwelling older adults in analyses of the bivariate analyses and the adjusted model. P.19 of 28, line 1-22.

„To test our hypothesis that the LSA-D is applicable in both urban and rural living environments, we calculated separate bivariate regressions for each subsample. The associations were similarly strong for functional mobility, self-rated health, balance confidence and history of falls. Although a significant association with age was only found for the urban population and limitations in ADL/iADL were not significant in either subsample, our findings generally correspond across both subsamples. This supports our notion that the LSA-D can be used for measurement of life-space mobility during the past four weeks in community dwelling older adults living in both urban and rural areas.

Contrary to our expectation results of the adjusted model calculated separately for each subsample revealed that limitations in ADL/iADL were only significantly associated with life-space mobility in urban areas. In contrast, shared living arrangements and history of falls were the only significant adjusted factors in rural areas. One possible explanation could be that life–space mobility achieved on one’s own ability’s is easier to maintain in urban areas with a more pronounced infrastructure. On the contrary a nearby social network may play a more important role for sustaining life–space mobility in rural areas where distances to services and social activities might be longer. This strengthened the importance of social resources on life-space mobility in rural areas. In this regard, Kusipar et al.[5] also found evidence for the importance of social support on life–space mobility in the Canadian Longitudinal Study of Aging. Due to the small sample size our results should be interpreted with caution and additional studies are needed to confirm the observed differences between urban and rural community-dwelling older adults found in our study. Thereby, future studies should continue to establish a theoretical and empirical basis for urban/rural life-space mobility.”

[5] Kuspinar A, Verschoor CP, Beauchamp MK, et al. Modifiable factors related to life-space mobility in community-dwelling older adults: results from the Canadian Longitudinal Study on Aging. *BMC Geriatr* 2020;20:35.

19. : “we tested psychometric properties.-.” Actually, only construct validity was tested, but no other psychometric property (e.g. test-retest, feasibility, responsiveness, etc.). This could also be mentioned as future research.

Response 19: Again, we would like to thank the reviewer for addressing this important point.

Action 19: In line with the reviewers first comment we changed the term psychometric properties throughout the whole text and now use the term construct validity instead.

We also named test-retest reliability and responsiveness as further psychometric properties of the LSA-D that should be evaluated in future studies. Discussion p.21 line 4-6 now says:

“Future studies should replicate our findings in a representative sample, including different subgroups of older adults and evaluate additional psychometric properties of the LSA-D as test-retest reliability and responsiveness.”

Reviewer 2:

1. Page 5 of 28, second paragraph: “like external control believes” – should “believes” be “beliefs” here?

Response 1: We thank the reviewer for spotting this point, we changed believes it into “beliefs” accordingly.

Action 1: In accordance to the reviewers comment we changed the sentences on p.5, line 10 as follows: “Additionally, psychological health factors like external control beliefs [20] and personal activity goals [21] influence life-space mobility.”

2. Later same paragraph: “activities an instrumental activities of daily living (ADL/iADL)” – “an” doesn’t seem to be the correct word to use here. Please re-word.

Response 2: Thanks to the reviewer for spotting this typo.

Action 2: We reworded the sentence and now it says on p. 5, line 12-14 :

“The construct validity of the LSA was commonly tested by relating the LSA to activities and instrumental activities of daily living (ADL/iADL) but also self-rated health and fears of falling.[22, 23, 24]”

3. There is a question about novelty of the study – please indicate whether the LSA has been validated in other countries for urban and rural community-dwelling older adults and why a validation for the German version is justified

Response 3: Thanks to the reviewer for this important remark. We added more information on the importance of the validation of the LSA in both urban and rural settings, which has also been pointed out in the original validation study of Baker et al.. Now the revised sentence reads as follows:

Action 3: Page 5 of 28, line 14-20: “Moreover, as pointed out by Baker et al.[12], there is a need to validate the LSA for urban and rural settings. Recently published studies also indicate environmental factors such as distance to services or quality of streets and sidewalks that differ between urban and rural settings might influence life-space mobility by reducing or maximising the opportunities to move oneself independently outdoors and participate in social activities.[25] As part of validity testing, the LSA has been translated into multiple languages such as Chinese[26], French[24], Spanish[22], Swedish[27] or Danish[28].”

[12] Baker PS, Bodner EV, Allman RM. Measuring life-space mobility in community-dwelling older adults. *J Am Geriatr Soc.* 2003 Nov;51(11):1610-4. doi: 10.1046/j.1532-5415.2003.51512.x. PMID: 14687391.

[22] Curcio CL, Alvarado BE, Gomez F, et al. Life-Space Assessment scale to assess mobility: validation in Latin American older women and men. *Aging clinical and experimental research* 2013;25:553-60

[24] Auger C, Demers L, Gelinat I, et al. Development of a French-Canadian version of the Life-Space Assessment (LSA-F): content validity, reliability and applicability for power mobility

[25] Miyashita T, Tadaka E, Arimoto A. Cross-sectional study of individual and environmental factors associated with life-space mobility among community-dwelling independent older people. *Environmental Health and Preventive Medicine* 2021;26:9.

[26] Tseng YC, Gau BS, Lou MF. Validation of the Chinese version of the Life-Space Assessment in community-dwelling older adults. *Geriatr Nurs* 2020;41:381-6.

[27] Fristedt S, Kammerlind AS, Bravell ME, et al. Concurrent validity of the Swedish version of the life-space assessment questionnaire. *BMC Geriatr* 2016;16:181.

[28] Pedersen MM, Kjaer-Sorensen P, Midtgaard J, et al. A Danish version of the life-space assessment (LSA-DK) - translation, content validity and cultural adaptation using cognitive

4. It is indicated participants gave consent to participate, but please also add whether the study was approved by an ethics board.

Response 4: We would like to thank the reviewer for this comment. We also see the importance of

approval of the study by an ethics board. To make this transparent for the reader we already referred to the approval of the local Ethics Committee of the Charité – Universitätsmedizin Berlin (EA2/124/19) on p.6 of 28 line 25.

Action 4: No action required.

5. Top of page 9 of 28: For the scoring “(0. Mobility within the bedroom, 1. rooms inside the home besides the bedroom, 2. area outside the house, 3. neighbourhood, 4. town or city lived in, outside of town or city lived in). Is this correct, or is there a score missing for the last category “outside of town or city lived in”?

Response 5: We want to thank the reviewer for finding this missing score. Indeed, we missed to number the last life-space correctly with 5. To make the different assessed life-spaces clearer we separated each numbered life-space with a semicolon and added the missing number 5. To the last category of life-space.

Action 5: Page 9 of 28, line 7-10: “The LSA consists of a questionnaire on five different life-spaces capturing six possible levels of life-space (0. mobility within the bedroom; 1. rooms inside the home besides the bedroom; 2. area outside the house; 3. neighbourhood; 4. town or city lived in; 5. outside of town or city lived in).”

6. Same page, last paragraph: Please define the abbreviation “TUG” the first time you use it.

Response 6: We would like to thank the reviewer again for this comment.

Action 6: We added the description of the abbreviation in the measurement section. Before this the abbreviation was defined in the section “Aims and Hypotheses” but we also changed that in response to one of the first reviewer’s comments. Now we use the abbreviation the first time in the measurement section under the subheading “Secondary outcome measures” and the new sentence on p.10 of 28, line 8 now reads as follows:

“The “Timed-Up&Go-Test” TUG is one of the most frequently used measures of balance and functional mobility in older adults and is a recommended tool for geriatric assessment.[39].”

[39] Turner G, Clegg A. Best practice guidelines for the management of frailty: a British Geriatrics Society, Age UK and Royal College of General Practitioners report. Age and Ageing 2014;43:744-7.

7. Last sentence on this page “in a comfortable self-selected speed” – change “in” to “at”

Response 7: Thanks again to the reviewer for finding this typo.

Action 7: We corrected the mistake as the reviewer has suggested (p. 10 of 28; line 12).

8. Page 10, first sentence: “Higher TUG times are associated with stronger mobility and ADL restrictions” – I think “stronger mobility” is incorrect here. Should this be “impaired mobility”?

Response 8: Thanks to the reviewer for pointing out that our description of the TUG lead to misunderstandings. We described that higher TUG times are associated with stronger mobility restrictions. Impaired mobility seems to be the better term here.

Action 8: Page 10 of 28; line 12-13: As the reviewer suggested we changed the term “stronger mobility and ADL restrictions” to “impaired mobility”:

“During performance of the TUG, time (in seconds) is taken for rising up from a standardized chair, walking three metres, turning around, walking back and sitting down again at a comfortable self-selected speed.[40] Higher TUG times are associated with impaired mobility.[41, 42] “

[40] Podsiadlo D, Richardson S. The timed "Up & Go": a test of basic functional mobility for frail

elderly persons. Journal of the American Geriatrics Society 1991;39:142-8.

[41] Lin MR, Hwang HF, Hu MH, et al. Psychometric comparisons of the timed up and go, one-leg stand, functional reach, and Tinetti balance measures in community-dwelling older people. Journal of the American Geriatrics Society 2004;52:1343-8.

[42] Donoghue OA, Savva GM, Cronin H, et al. Using timed up and go and usual gait speed to predict incident disability in daily activities among community-dwelling adults aged 65 and older. Archives of physical medicine and rehabilitation 2014;95:1954-61.

9. End of second paragraph on this page: Change “criteria’s” to “criteria”

Response 9: Thanks to the reviewer for spotting this typo.

Action 9: We corrected this mistake in the referred manner.

P. 10 of 28; line 18-20:

“Participants were accounted to have a history of falls if they had fallen at least one time in the past six months using the criteria of the “Frailty and Injuries: Cooperative Studies of Intervention Techniques” to define a fall.[45]”

[45] Buchner DM, Hornbrook MC, Kutner NG, et al. Development of the common data base for the FICSIT trials. Journal of the American Geriatrics Society 1993;41:297-308.

10. Table 1: Could the authors add a column on urban vs. rural differences for the assessed variables (i.e. p-value?).

Response 10: Thank you for this comment; we added an indication of significance to the table. To make the differences clearer we marked significant differences in bold letters and classified the differences in*(p< .05) and ** (p<.001).

Action 10: The notes of table 1 now say:

Note: Live-Space-Assessment-Deutsch (LSA-D); Activities of Daily Living (ADL); Instrumented Activities of Daily Living (iADL); Timed-Up &Go-Test (TUG); Numbers in brackets report number of missing values; Significant differences between subsamples are highlighted in bolt letters and marked with *(p< .05) and ** (p<.001)

11. Figure: “LS-C composite life-space” is presented in the figure legend, but this does not appear in the figure

Response 11: Many thanks to the reviewer for noticing this mistake in the figure description. It was a simple mistake in the creation of the figure description.

Action 11: We changed the figure description accordingly and removed the LS-C = composite life-space from the figure description.

All new included references in the manuscript:

[25] Miyashita T, Tadaka E, Arimoto A. Cross-sectional study of individual and environmental factors associated with life-space mobility among community-dwelling independent older people. Environmental Health and Preventive Medicine 2021;26:9.

[29] Hauer K, Ullrich P, Heldmann P, et al. Psychometric Properties of the Proxy-Reported Life-Space Assessment in Institutionalized Settings (LSA-IS-Proxy) for Older Persons with and without Cognitive

Impairment. International journal of environmental research and public health 2021;18:3872.
 [5] Einig K, Zaspel-Heisters B. Das System Zentraler Orte in Deutschland. In: Flex F, Greiving S, eds. Neuaufstellung des Zentrale-Orte Konzepts in Nordrhein-Westfalen. Hannover: Verlag des ARL 2016.
 [31] Auais M, Alvarado B, Guerra R, et al. Fear of falling and its association with life-space mobility of older adults: a cross-sectional analysis using data from five international sites. Age Ageing 2017;46:459-65.

VERSION 2 – REVIEW

REVIEWER	Werner, C Heidelberg University, Centre for Geriatric Medicine
REVIEW RETURNED	10-Jun-2021

GENERAL COMMENTS	I appreciate the thorough and well-done revisions by the authors, which have significantly improved the manuscript. All my comments were adequately addressed. I do not have any further comments and can recommend the acceptance of the manuscript for publication.
---

REVIEWER	Chilibeck, Philip University of Saskatchewan
REVIEW RETURNED	21-Jun-2021

GENERAL COMMENTS	The authors have adequately revised the manuscript. I have the following minor suggestions for revision: Page 6, line 15: Change “hypothesizes” to “hypotheses” Page 9, line 6: “some towns missed to provide services” – change to “some towns did not provide services...” Line 17: “medical professionals were consulted by any uncertainty...” – change “by” to “for” Page 21, line 18: Change “ability’s” to “abilities” Line 25: Change “he” to “the” Page 23, line 19: Change “as” to “such as”
---